# The Differential Heritability of Social Adjustment by Sex

**DOI:** 10.3390/ijerph18020621

**Published:** 2021-01-13

**Authors:** Chisato Hayashi, Soshiro Ogata, Haruka Tanaka, Kazuo Hayakawa

**Affiliations:** 1Research Institute of Nursing Care for People and Community, University of Hyogo, 13-71, Kitaoji-cho, Akashi, Hyogo 673-8588, Japan; 2Department of Preventive Medicine and Epidemiology, Natl. Cardiovascular Center, 5-7-1 Fujishirodai, Suita, Osaka 564-8565, Japan; s_ogata@ncvc.go.jp; 3School of Health Science, Graduate School of Medicine, Nagoya University, 1-1-20, Daiko-Minami, Higashi-ku, Nagoya City 461-8673, Aichi, Japan; haruka.tanaka@met.nagoya-u.ac.jp; 4Division of Health Science, Graduate School of Medicine, Osaka University, 1-7 Yamadaoka, Suita, Osaka 565-0871, Japan; hayakawa@sahs.med.osaka-u.ac.jp

**Keywords:** school adaptation, twin, sex difference, heritability

## Abstract

Sex differences in social adjustment are frequently observed; however, there has been very little research on adaptability in the individual and social domains. The aim of this study was to investigate the sex difference in social abilities, such as high self-appeal, sociability, school adaptation, and home adaptation between school-age males and females. The sample for this study included both same-sex and opposite-sex twin pairs: a total of 467 twin pairs. We classified them into three groups: a group of those in lower classes of elementary school, a group of those in higher classes of elementary school, and a group of those in junior high school. The heritability of school adaptation was estimated to be 95% in males and 54% in females in the junior high school group. The full sex-limitation model showed a better fit in this group, and this means that a qualitative genetic difference exists. For school adaptation, there was no sex difference in lower elementary school classes; however, a quantitative difference appeared in higher classes of elementary school. Moreover, a qualitative difference appeared in junior high school. From this research, it became clear that sex differences in heritability exist for school adaptation, and there was a marked increase from the elementary school children to the junior high school children.

## 1. Introduction

The number of absentee students at elementary and junior high schools in the 2019 academic year was 181,272, the highest number ever in Japan [1]. School absenteeism is understood as a “separation anxiety disorder” stemming from anxiety over mother–infant separation [2,3,4]. A twin study reported that shared environmental effects (40%) were found to play a moderate role in relation to the symptoms of separation anxiety, and the parameter approached significance among 11 to 13-year-old males [5].

Twin studies are the mainstay of behavioral genetics and serve as a crucial tool in establishing the heritability of a phenotype [6]. In research on twins, many behavioral genetic studies have dealt with the subject of personality, such as tendencies for extroversion/nervousness [7] and the Big Five Questionnaire, neuroticism (N), extraversion (E), openness to experience (O), agreeableness (A), and conscientiousness (C) [8,9,10,11,12,13,14]. These studies found significant stable genetic effects (about 30–50%) and nonshared environmental effects (about 50–70%). According to the meta-analysis, genetics contribute an average of 40% to individual differences in personality [15].

Recent twin studies have also suggested the persistent influence of X-linked genes on cognition and social behavior problems from early to middle childhood. It is important to stratify genetic analyses of behaviors by gender [16]. On the other hand, there is evidence that neither quantitative nor qualitative sex differences in anxiety, depression, and somatic complaints exist [17], and that only quantitative sex differences in alcohol dependence and binge eating exist in adolescents [18]. Haworth et al. (2010) suggested that the developmental increase in the heritability of general cognitive ability lies in genotype–environment correlation: as children grow up, they increasingly select, modify, and even create experiences in part on the basis of their genetic propensities [19].

Social adjustment is an insufficient explanation of sex development, and there is now increasing recognition that biological predispositions play a role in sex-related characteristics as in other psychological traits [20]. In a population-based rather than clinical study, there was evidence that social adjustment was significantly worse in males and had strong heritability, but there was no evidence to support significant differences in genetic effects between males and females [21]. Females had significantly better social adaptation than males and were suggested to show an influence of a locus expressed only from a parental X chromosome [22].

There has been very little research on the differential heritability of school-age children’s social adjustment and on adaptability in the individual and social domains, including adaptability to social, school, and home environments by sex. A twin study reported that genetic factors are important for explaining adolescent behavioral problems, especially for girls, while shared environmental influences cannot be ignored for boys [23]. They reported that sex differences in heritability exist for behavioral/emotional problems.

The aim of this study was to conduct sex-limitation modeling utilizing a study design to investigate the influence of sex in regard to social adjustment. In this study, we hypothesized that variance in social abilities, such as high self-appeal (that is, making one’s presence known to others), sociability, school adaptation, and home adaptation would have a significant genetic component, and that genetic effects would differ between school-age males and females.

## 2. Materials and Methods

### 2.1. Participants

In total, 467 twin pairs aged 7–15 years old (from 1st year of elementary school to 3rd year of junior high school) were included. The mean age of twins was 10.8 ± 2.3 years old. The sample for this study included both same-sex and opposite-sex twin pairs, allowing the investigation of sex difference in social stability: 101 dizygotic opposite-sex (DZos), 87 DZ male (DZm), 82 DZ female (DZf), 101 monozygotic male (MZm), 96 MZ female (MZf).

### 2.2. Procedure

We sent questionnaires to 2733 members of the Twin Mothers’ Club who gave birth to twins between 2 April 1988 and 1 April 1997. We received 1428 responses to the questionnaire (52% response rate). The Twin Mothers’ Club is the largest organization for mothers of twins and higher order multiple births in Japan. In the present study, we sent a questionnaire in 2006 as a follow-up study to 958 mothers who were the subjects of the study in 1999. As a result, 516 respondents returned the questionnaire (53.9%). In our previous study, 14 pairs of twins were excluded from the study because of cerebral palsy, cleft palate, autism or Down’s syndrome, to remove influential outliers [24]. The reason these twin pairs were excluded is that these conditions affect social development, thereby introducing the possibility of influential outliers. Removed pairs totaled 33 pairs aged 16 years old (high school age) and 5 pairs aged 19.

We classified the 467 twin pairs into three groups: a group of those in lower elementary school (7–9 years old, 152 pairs), a group of those in higher elementary school classes (10–12 years old, 192 pairs), and a group of those in junior high school (13–15 years old, 123 pairs).

### 2.3. Human Subjects’ Approval Statement

We mailed a summary of the research and obtained agreement to participate in the present study if the respondents answered voluntarily and handed in the unsigned self-reported questionnaires. The study had the approval of the Ethics Committee of the School of Medicine, Osaka University (No. 466). The funders had no role in study design, data collection and analysis, decision to publish, or preparation of the manuscript.

### 2.4. Instruments

A population-based sample of twins aged 7–15 years and their mothers responded to questionnaire items, including those related to four types of social adjustment (high self-appeal, sociability, school adaptation, and home adaptation). The four social measures were measured as “stability of social human relationships” in “TS style: Infant and child personality diagnostic test”, a mother-reporting test of child personality that can be answered easily and in a short time [25]. Regarding reliability, the result of the test–retest method was r = 0.79~0.92, the result of the split-half method was rtt = 0.59~0.80, and Cronbach’s alpha for measures of stability was 0.86. In this study, we picked four factors: high self-appeal, sociability, school adaptation, and home adaptation. The four factors were evaluated using 12 questions, and each question was answered on a 2-point scale (yes/no).

High self-appeal was evaluated using the following questions: “Is he/she often very selfish?”, “Is he/she often rude?”, “Does he/she often make excuses?”, “Does he/she always have his/her own way?”, “Does he/she focus too much on chatting about himself/herself?”. Sociability was evaluated using the following questions: “Is he/she very shy in public?”, “Does he/she have few friends in general?”, “Does he/she like to be alone more than with a number of children?”, “Does he/she like being at home and not like going out very much?”. Home adaptation was evaluated using the following questions: “Does he/she care about his/her parents’ going away very much and is he/she inquisitive about their whereabouts?”, ‘‘Does he/she seldom talk about events at school?”, “Does he/she dislike anyone in the household very much?”, “Does he/she cling to his/her parents?”, “Does he/she seem extremely frightened when his/her parents scold him/her?”. School adaptation was evaluated using the following questions: “Is he/she sometimes unwilling to go to school?”, “Is he/she quiet at school?”, “Does he/she have few friends?”, ”Does he/she often appear to have physical disorders such as stomachaches and headaches when he/she goes to school?”, “Does he/she not get on well with the teacher at school?”, “Is he/she often teased at school?”.

We used the question, “Are the twins as alike as two peas in a pod?” for zygosity classification. Twin pairs for whom the answer to this question was “Yes” were considered to be monozygotic (MZ), and those for whom the answer was “No” were considered to be dizygotic (DZ). Their zygosity was previously identified by several genetic markers. Previous studies indicate that more than 90% of twins were diagnosed correctly using the alikeness question [26,27,28].

### 2.5. Data Analysis

#### 2.5.1. Model Fitting of Twin Study Methods

We conducted model-fitting analyses to estimate the relative contribution of genetic and environmental factors to the variation in social stability using the twin study method. Monozygotic (MZ) twins share 100% of their genes, whereas dizygotic (DZ) twins share approximately 50% of their genes. If within-twin pair correlations are higher than what would be predicted from the heritability trait (A), shared environment effects (C) are implicated. The remaining environmental variance is therefore a nonshared environmental effect (E) or measurement error. The present study tested for three sex differences: (a) qualitative, (b) quantitative, and (c) variance differences between males and females [29,30]. To investigate sex differences, the present study performed comparisons among models that allowed sex differences (*full sex-limitation model*) and models that did not allow sex differences (the *common effects model*, *scalar model*, and *null model*) using data divided into MZm, MZf, DZm, DZf, and DZos twin pairs, which were suitable for investigating sex differences in genetic and environmental components [29,30,31].

#### 2.5.2. Qualitative Sex Differences with Sex-Limitation Model Fitting

Qualitative sex differences were tested to investigate whether the same genetic and environmental factors influenced individual social differences in the phenotype for males and females (Figure 1) by comparing the model fit of the *full sex-limitation model* and *common effects model*. To allow for the investigation of sex differences, a full sex-limitation model estimated all parameters separately for males and females.

#### 2.5.3. Quantitative Sex Differences with Sex-Limitation Model Fitting

Quantitative sex differences were tested to investigate whether the same genetic and environmental factors influenced, but to a different degree, the influence of A, C, or E on social measures, by comparing the model fit of the *common effects model* and *scalar effects model*. It was noted that qualitative and quantitative sex differences were not associated with any observed mean sex differences. To allow for the investigation of quantitative sex differences, a common effects model constrained the opposite-sex twins’ additive genetic-relatedness coefficient (r*_g_*) to equal 0.5 and shared environment-relatedness coefficient (rc) to equal 1.0, but estimated the path coefficients of A, C, and, E parameters separately for males and females. To allow neither quantitative nor qualitative sex differences, a scalar model constrained the opposite-sex twins’ additive genetic-relatedness coefficient (r*_g_*) to equal 0.5 and shared environment-relatedness coefficient (rc) to equal 1.0, and estimated the path coefficients of A, C, and, E parameters equally for males and females. A null model did not allow qualitative, quantitative, or variance differences between males and females by constraining the opposite-sex twins’ r*_g_* to 0.5 and rc to 1.0, equating the path coefficients of A, C, and E parameters for males and females, and equating phenotypic variance for males and females.

#### 2.5.4. Interpretation of the Results of Model Comparisons

Interpretation of the results of model comparisons was as follows [30,31]: Differences in fit between the full sex-limitation model and a common effects model showed qualitative sex differences. In addition, differences in fit between the common effects model and scalar model showed quantitative sex differences. Furthermore, differences in fit between the scalar model and null model showed no variance differences between males and females. Model comparison was performed based on Akaike information criterion (AIC) and likelihood ratio tests [32]. The model with the lowest AIC indicated the most parsimonious (preferred) model. Likelihood ratio tests showed significant *p*-values in model comparisons, indicating significant differences in fit between models. A difference in AIC between two models of 2 or less provides equivalent support for both models. On the other hand, a difference in AIC between two models of 3 indicates that the lower AIC model has considerably more support, and a difference in AIC between two models of more than 10 indicates that the lower AIC model is a substantially better fit compared to the higher AIC model [32].

All statistical analyses were performed using R 3.0.3 [33], and twin modeling was performed using the OpenMx package [34]. All parameters, their 95% confidence intervals (CI), and *p*-values were obtained with adjustment for age via a residual method [35]. All tests conducted were two-tailed, with the type I error set at 0.05.

## 3. Results

### 3.1. Descriptive Statistics

In Table 1, the mean and standard deviation for measure were presented. One-way analysis indicated significant differences in high self-appeal (*p* < 0.001), sociability (*p* = 0.023), and home adaptation (*p* = 0.043). There were significant differences between the lower classes and junior high school in high self-appeal and between the higher classes and junior high school with multiple comparisons (Bonferroni). Table 2 shows the bivariate correlation matrix (Pearson). At the bivariate level, there was a significant correlation between sociability and school adaptation (*p* < 0.001), between high self-appeal and home adaptation (*p* < 0.001), and between sociability and home adaptation (*p* < 0.001). High-self appeal was not related to sociability and school adaptation. Table 3 shows intraclass correlations by zygosity and sex. The total group included MZ, DZ, DZ same-sex (DZss), and DZ opposite-sex (DZos) groups, as well as subgroups among same-sex pairs, males (DZm), and females (DZf). In all measures, correlations were higher in MZ pairs than in DZ pairs. If the difference in intraclass correlation between the MZ and DZ groups was more than double, the phenotype was considered to have a substantial genetic influence. Where a correlation was higher for DZss twins as compared to DZos twins, a specific gene was assumed to be playing a role.

For high self-appeal, correlations between MZ twins were higher than those between DZ twins, suggesting a genetic influence. In the lower classes and higher classes of elementary school, correlations between DZ twins were less than half those between MZ twins, suggesting two assumptions. First, there would be some specific shared environmental factors which cause correlations between MZ pairs. Second, a nonadditive genetic effect may exist. In the junior high school group, the correlations between MZ twins and DZ twins were nearly equal; thus, some peculiar twin situation may exist. For sociability, in the lower and higher elementary school classes, the correlations between DZ twins were more than half the correlations between MZ twins, suggesting a shared environmental contribution. In the junior high school group, correlations between DZ twins were less than half those of DZ twins. For school adaptation, in the lower classes and junior school, correlations between DZ twins were less than half those between MZ twins. In higher elementary school classes, correlations between DZ twins were more than half those between MZ twins. For home adaptation, correlations between DZ twins were nearly equal to those between MZ twins for high school adaptation scores, suggesting strong effects of a shared environment.

### 3.2. Sex-Limitation Model Fitting

There was a tendency for correlations between DZss twins to be higher than those between DZos twins for most measures except sociability in the lower elementary school classes, and for school adaptation and home adaptation in the junior high school group. Therefore, a full sex-limitation model was examined in which there were different degrees of genetic effects for males and females (quantitative differences), as well as different genes influencing (qualitative differences) all four measures in each group (lower classes and higher classes of elementary school and junior high school). Second, we tested a sex-limitation model in which only different genes played an influencing role (qualitative differences), by comparing a full sex-limitation model and a common effects sex-limitation model. If the common effects sex-limitation model was significantly different compared to the full sex-limitation model, this suggested a qualitative sex difference. Third, we tested a scalar sex-limitation model which specified the same degrees of genetic effects and the same genes for males and females (quantitative differences) by comparison between a common effects model and a scalar effects model. If the scalar effects model was significantly different compared with the common effects model, this suggested a quantitative sex difference. Finally, we tested a null model which indicated no qualitative or quantitative differences for all four measures of as high self-appeal scores, sociability scores, school adaptation scores, and home adaptation scores.

#### 3.2.1. Qualitative Sex Differences

Table 4 shows a comparison of full sex-limitation models and common effects models to consider qualitative sex differences. For three measures (high self-appeal, sociability, and home adaptation) in the groups of lower and higher elementary school classes and junior high school, the common effects model was not fitted well. That is, there were not qualitative sex differences, and the same genetic and environmental factors influenced individual social differences in the phenotype for males and females. In the junior high school group, for school adaptation only, compared to *the common effect model*, *the full sex-limitation model* fitted the data well (*p* < 0.001, AIC = 435.27, −2LL = 909.27, df = 237). Namely, both qualitative (different genes influencing males and females) and quantitative (different degrees of genetic and environmental effects) genetic sex differences exist in the junior high school group.

#### 3.2.2. Quantitative Sex Differences

Table 5 shows a comparison of three sex-limitation models: a common effects model, a scalar effects model, and a null model to consider quantitative sex difference. For high self-appeal in junior high school, compared to the scalar effects model, the common effects model fitted the data well (*p* = 0.011, AIC = 522.22, −2LL = 998.23, df = 238). This suggests that quantitative sex differences exist. The null model did not provide an adequate fit for high self-appeal data.

In sociability, null models best fitted data of lower elementary classes, higher classes and junior high school. This suggests that no sex difference existed. In school adaptation, compared with the scalar effects model, the common effects model fitted the data well in the higher elementary school classes (*p* < 0.001, AIC = 668.52, −2LL = 1420.53, df = 376) and junior high school group (*p* < 0.001, AIC = 445.48, −2LL = 921.48, df = 238). This suggests that quantitative sex differences exist. For home adaptation, the common effects model only fitted the data well (*p* < 0.01, AIC = 677.01, −2LL = 1429.01, df = 376) in the higher elementary school classes when compared to the scalar effects model. This suggests that quantitative sex differences exist.

In summary, for school adaptation, our findings suggest that there is a qualitative sex difference in junior high school adaptation and a qualitative sex difference in higher elementary school classes. The sex difference increased markedly from the higher elementary school classes to the junior high school group. For high self-appeal and home adaptation, the sex differences were small, and for sociability, there was no sex difference in all groups. The results of the full sex-limitation model and best fit model are reported in Table 6 (high self-appeal), Table 7 (sociability), Table 8 (school adaptation), and Table 9 (home adaptation).

For sociability no sex difference exists in all groups. In elementary school groups, 50% to 59% heritability was found, with shared environmental influences accounting for 21% to 32% and nonshared environmental influences accounting for 18% to 21% of variance. In junior high school, 73% heritability was found, with nonshared environmental influence accounting for 27% of variance. Heritability increased from 50% to 73% from lower elementary school classes to junior high school in males and females.

For school adaptation, in higher elementary school classes, sex differences exist. In males, 49% heritability was found, with shared environmental influences accounting for 44% and nonshared environmental influences accounting for little (7%) variance, while in females, 63% heritability was found, with nonshared environmental influences accounting for 36% of variance. Sex differences also exist in the junior high school group. In males, high heritability (95%) was found, with nonshared environmental influences accounting for little (5%) variance, while in females, 54% heritability was found, with nonshared environmental influences accounting for 46% of variance. Heritability in males decreased from 77% to 49% in lower to higher elementary school classes; it increased from 49% to 95% in higher elementary school classes to junior high school. Heritability in females decreased gradually from 77% to 63% to 54%.

For home adaptation, sex difference exists in only higher elementary school classes. In males, 35% heritability was found, with shared environmental influences accounting for 43% and nonshared environmental influences accounting for 23% of variance, and in females, 21% heritability was found, with shared environmental influences accounting for 69% and nonshared environmental influences accounting for 10% of variance.

## 4. Discussion

This study sought to investigate the sex difference of social adjustment in elementary school and junior high school children. Major influences on individual differences in high self-appeal in the elementary school groups included additive genetic and nonshared environmental factors. However, in the junior high school group, the major influence included a shared environment, particularly in males. This suggests that there is a special twin environment (the “twin situation”), a unique situation created by the presence of a sibling of the same age during development. Particularly, in junior high school, males develop secondary sexual characteristics. Initially, we expected that correlations between MZ twins in high self-appeal would be higher than those between DZ twins because they have a sense of rivalry with each other; however, the result was the opposite of what we expected. One possible explanation for this is that if siblings are more alike like MZ twins, less competitive spirit between each other will arise in adolescent males.

The heritability of school adaptation was estimated to be 95% in males and 54% in females in the junior high school group. The full sex-limitation model showed a better fit in this group, and this means a qualitative genetic difference exists. For school adaptation, there was no sex difference in lower elementary school classes; however, a quantitative difference appeared in higher classes of elementary school. Moreover, a qualitative difference appeared in junior high school. For secondary sexual characteristics, specific genetic expression would have effects on school adaptation in junior high school males. One possible explanation for this increase in heritability is gene–environment correlation or a changing relationship between genes and the brain; for example, the heritability of white matter volume increases through childhood to adolescence, mirroring the increase in heritability [36,37,38]. As additional factors, relationships with friends and a nonshared environment may affect school adaptation in females from higher classes of elementary school to junior high school. One reason for the high genetic effect on school adjustment among junior high school students may be the influence of the junior high school entrance exam. Although this study did not investigate whether twins attend the same school, more twins will likely attend separate schools from middle school, especially male and female twins. For example, males may go to a boys’ school and females to a girls’ school. This may have increased the difference between male and female twins in school adjustment, which may have affected the results.

For sociability, there was an increasing grade from lower elementary school classes to junior high school, and the variance of A became greater, while the variance of C became 0 (AE model). Some possible explanations for the increase in heritability were reported. First, active or evocative gene–environment correlation may occur. Second, the genetic influence increases because environmental variability decreases. Ando (2014) suggested that absence of a shared environmental effect in personality is involved in qualitative characteristics that cannot return to quantitative characteristics. For example, regarding the school record, if children show an increased quantity of learning, their school records would improve; however, an increase in quantity of experience in society would not directly lead to increased sociability [39].

For home adaptation, a sex difference appeared in the higher elementary school class classes. Notably, the variance of C became large in females. This result also might suggest that the sibling environment—in this case, the special twin environment (“twin situation”) —affects home adaptation. In females, secondary sexual characteristics appear around the higher elementary school age, earlier than in males. On the other hand, in males, school adaptation might be greatly influenced by the twin relationship relating to secondary sexual characteristics, and in females, home adaptation might be greatly influenced.

This demonstrates the presence of qualitative and quantitative sex differences in school adaptation. In previous surveys, we did not encounter reports describing the causes of school absenteeism by sex, which is why there is a need to investigate the causes from this perspective. On the other hands, there is evidence of poorer social cognition among males than females [21]. Heritability of 68% was found, with shared environmental influences accounting for only 5% of the variance. There were, however, no significant differences in genetic effects between sexes. In addition, social cognition appears to be under considerable genetic influence in the population and shows significant sex differences [21].

From this research, it has become clear that when it comes to explaining social adjustment, the rate of explanation by genetic and environmental factors differs between sexes. In the case of junior high school males, the major presence of genetic factors (95%) suggests the possibility that an individual’s personality or temperament has a bearing on school absenteeism. Meanwhile, in the case of females, environmental factors have a large influence (46%), suggesting the possibility that relationships with friends and twins, in addition to the home environment, have a bearing on school absenteeism. The results of this research have the potential to contribute to the development of effective approaches for dealing with school adaptation in adolescents.

Twins are often compared with those around them from birth, and self-appeal or adaptation to group styles may be different from that of singletons. This research helps us to consider not only genetic and environmental variances in social adjustment, but also how schoolchildren and adolescents adapt to their environment.

## 5. Limitations

There are several limitations in our study. First, this study used a parent-reporting questionnaire on twins’ social ability. Parent ratings of children’s personality are affected by contrast effects that exaggerate estimates of genetic influence in twin studies. Second, while the legitimacy and reliability of the scales employed in this research have been verified, the study’s scope is limited by the fact that its scales were created in Japan and have not been used in international comparisons. In addition, the zygosity questionnaire consisted of one question. Third, it should be noted that as a result of the small sample size and wide age range (7–15 years old), the power to conduct model comparisons and parameter estimation was limited. For example, the present study does not have the statistical power to detect a model for each age of twins in order to distinguish differences for the assessment of age differences in the model fit. Therefore, to strengthen the present findings, replication with a larger sample size will be necessary in the future. Last, because we did not investigate whether twins attend the same school, we could not consider the impact of this on school adjustment.

## 6. Conclusions

For school adaptation, only in the junior high school group do both qualitative (different genes influencing males and females) and quantitative (different degrees of genetic and environmental effects) genetic sex differences exist. In addition, for school adaptation, in higher elementary school classes, only quantitative sex difference exists. For high self- appeal, in the junior high school group, quantitative genetic sex difference exists. For home adaptation, in higher elementary school classes, quantitative genetic sex difference exists. For sociability, no genetic sex difference exists.

This study found a small but significant genetic sex difference in social adjustment from childhood to adolescence. First, this finding suggests that adaptation to elementary school or junior high school has genetic sex difference. Second, there is also difference between sexes in adaptation to home. Particularly, female schoolchildren in the upper grade and male adolescents in junior high school start to develop secondary sex characteristics.

## Figures and Tables

**Figure 1 ijerph-18-00621-f001:**
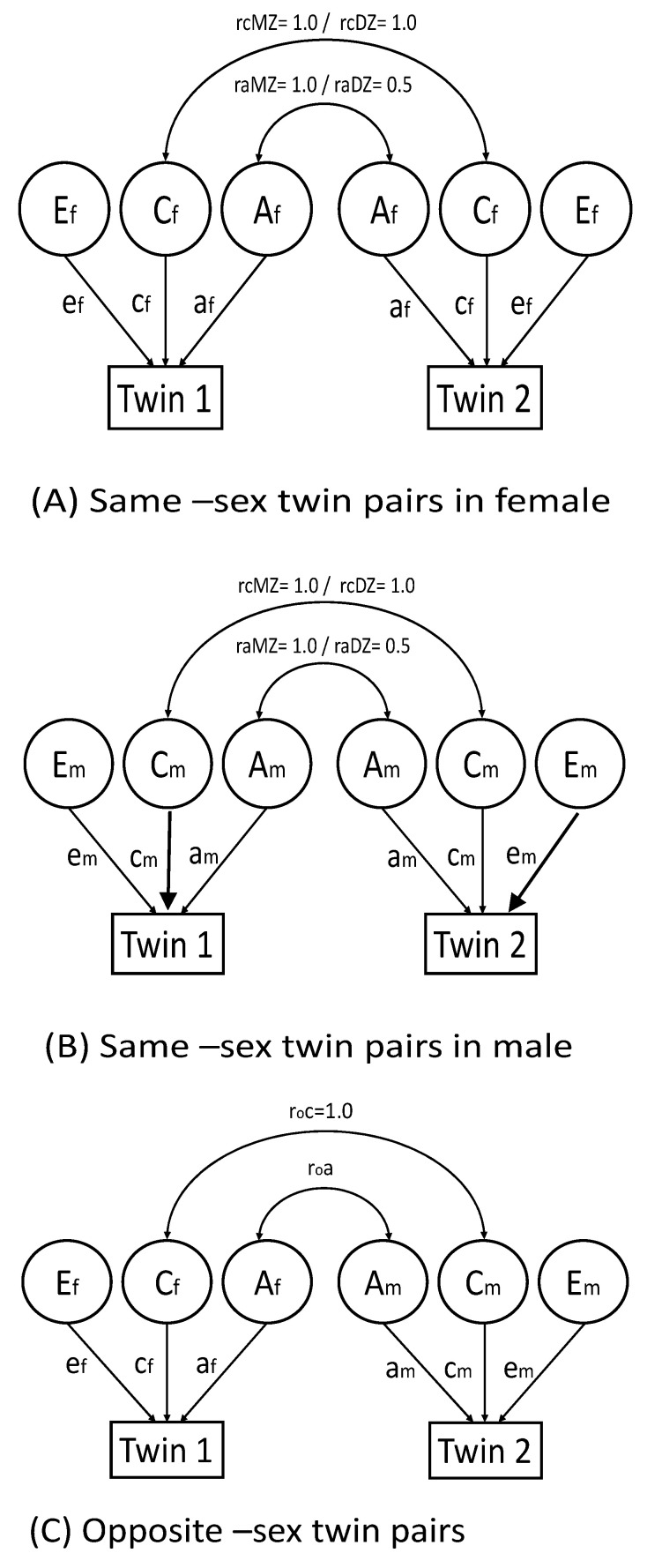
(**A**) Same-sex twin pairs in females, (**B**) same-sex twin pairs in males, (**C**) opposite-sex twin pairs. Abbreviations: MZ, monozygotic; DZ, dizygotic; Af, additive genetic factors for female sex; Cf, shared environmental factors for female sex; Ef, nonshared environmental factors for female sex; Am, additive genetic factors for male sex; Cm, shared environmental factors for male sex; Em, nonshared environmental factors for male sex; af, cf, and ef were path coefficients for females; am, cm, and em were path coefficients for males; af, cf, and ef were path coefficients for female sex; am, cm, and em were path coefficients for male sex; raMZ, additive genetic correlation between monozygotic twins; raDZ, additive genetic correlation between same-sex dizygotic twins; roc = shared environmental correlation between opposite-sex twins; roa = additive genetic correlation between opposite-sex twins. Rectangles represent observed variables, and circles represent latent variables. Full sex-limitation models estimated all parameters separately in males and females. Common effect models estimated parameters separately in males and females, fixing roa = 0.5. Scalar models estimated parameters, fixing roa = 0.5, af = am, cf = cm, and ef = em. Null models estimated parameters, fixing roa = 0.5, af = am, cf = cm, ef = em, phenotypic variance of females = phenotypic variance of males.

**Table 1 ijerph-18-00621-t001:** The mean and standard deviation for measures: one-way analysis (N = 467).

Measure	The Lower Classes (N = 152)	The Higher Classes (N = 192)	Junior High School (N = 123)	*p*-Value	Multiple Comparison (Bonferroni)
	Mean ± SD	Mean ± SD	Mean ± SD		
(1) High self-appeal	3.35 ± 2.32	3.10 ± 2.29	1.96 ± 1.69	<0.001	Between lower classes and junior high school; between higher classes and junior high school
(2) Sociability	2.57 ± 2.40	2.62 ± 2.49	3.30 ± 2.54	0.023	n.s.
(3) School adaptation	1.28 ± 1.54	1.27 ± 1.73	1.65 ± 1.63	0.111	n.s.
(4) Home adaptation	2.73 ± 1.67	2.56 ± 1.88	2.16 ± 1.78	0.043	n.s.

**Table 2 ijerph-18-00621-t002:** Bivariate correlation matrix: Pearson (N = 467).

Measure	(1)	(2)	(3)	(4)
	*r*	*r*	*r*	*r*
(1) High self-appeal	1	−0.094	0.065	0.432 **
(2) Sociability		1	0.707 **	0.210 **
(3) School adaptation			1	0.330
(4) Home adaptation				1

Abbreviations: ** *p* < 0.001.

**Table 3 ijerph-18-00621-t003:** Intraclass correlations of four measures for twins by zygosity and sex (group difference).

Zygosity and Sex (N = 467) Intraclass Correlations					
	MZm	MZf	DZm	DZf	DZos
	(N = 101)	(N = 96)	(N = 87)	(N = 82)	(N = 101)
	*r*	*r*	*r*	*r*	*r*
Measure					
High self-appeal					
Lower classes	0.71	0.55	0.18	0.51	0.21
Higher classes	0.59	0.66	0.17	0.34	0.10
Junior high school	0.36	0.14	0.60	0.18	0.16
Sociability					
Lower classes	0.89	0.81	0.56	0.41	0.62
Higher classes	0.74	0.88	0.68	0.28	0.41
Junior high school	0.86	0.72	0.22	0.37	0.27
School adaptation					
Lower classes	0.91	0.66	0.22	0.62	0.27
Higher classes	0.9	0.77	0.77	0.37	0.37
Junior high school	0.9	0.51	0.16	0.1	0.78
Home adaptation					
Lower classes	0.68	0.79	0.83	0.57	0.52
Higher classes	0.82	0.91	0.56	0.79	0.55
Junior high school	0.72	0.88	0.66	0.61	0.70

Abbreviations: MZm = monozygotic male; MZf = monozygotic female; DZm = dizygotic male; DZf = dizygotic female; DZos = dizygotic opposite sex pairs.

**Table 4 ijerph-18-00621-t004:** Comparison of full sex-limitation models and common effects models to consider qualitative sex differences.

Measure	Model	−2LL	df	diffLL	diffdf	*p*-Value	AIC
High self-appeal							
Lower elementary school classes	Full sex-limitation	1421.39	295				831.39
	Common effects	1421.39	296	<0.00	1	0.996	829.39
Higher elementary school classes	Full sex-limitation	1774.76	375				1024.76
	Common effects	1774.95	376	0.19	1	0.663	1022.95
Junior high school	Full sex-limitation	998.06	237				524.06
	Common effects	998.23	238	0.16	1	0.686	522.23
Sociability							
Lower elementary school classes	Full sex-limitation	1312.45	295				722.49
	Common effects	1314.05	296	1.56	1	0.212	722.05
Higher elementary school classes	Full sex-limitation	1704.8	375				954.8
	Common effects	1705.13	376	0.32	1	0.570	953.13
Junior high school	Full sex-limitation	1112.81	237				638.81
	Common effects	1113.09	238	0.28	1	0.596	637.09
School adaptation							
Lower elementary school classes	Full sex-limitation	1114.44	295				524.44
	Common effects	1114.53	296	0.09	1	0.768	522.53
Higher elementary school classes	Full sex-limitation	1419.53	375				669.53
	Common effects	1420.53	376	1.00	1	0.997	668.52
Junior high school	Full sex-limitation	909.27	237				435.27
	Common effects	921.48	238	12.21	1	<0.001	445.48
Home adaptation							
Lower elementary school classes	Full sex-limitation	1110.83	295				520.83
	Common effects	1110.84	296	0.01	1	0.931	518.84
Higher elementary school classes	Full sex-limitation	1428.99	375				678.99
	Common effects	1429.01	376	0.02	1	0.881	677.01
Junior high school	Full sex-limitation	881.64	237				407.64
	Common effects	884.62	238	2.98	1	0.084	408.62

Abbreviations: −2LL = −log-likelihood; df = degrees of freedom; diffLL = difference in likelihood; *p*-value = associated with differences in likelihood ratio between each full sex-limitation model and common effects model; AIC = Akaike’s information criterion. The *p*-value shows no significant differences in likelihood between the full sex-limitation model and common effects model.

**Table 5 ijerph-18-00621-t005:** Comparison of common effects, scalar effects models and null models to consider quantitative sex difference.

Measure	Model	−2LL	df	diffLL	diffdf	*p*-Value	AIC
High self-appeal							
Lower elementary school classes	Common effects	1421.39	296				829.39
	Scalar effects	1425.82	299	4.43	3	0.219	827.82
	**Null**	**1425.82**	**300**	**0.00**	**1**	**0.351**	**825.82**
Higher elementary school classes	Common effects	1774.95	376				1022.95
	Scalar effects	1780.80	379	5.85	3	0.119	1022.8
	**Null**	**1780.80**	**380**	**0.00**	**1**	**0.211**	**1020.8**
Junior high school	**Common effects**	**998.23**	**238**				**522.22**
	Scalar effects	1009.30	241	11.07	0.011		527.3
	Null	1009.30	242	0.00	1	0.026	525.3
Sociability							
Lower elementary school classes	Common effects	1314.05	296				722.05
	Scalar effects	1316.04	299	1.99	3	0.574	718.04
	**Null**	**1316.04**	**300**	**0.00**	**1**	**0.737**	**716.04**
Higher elementary school classes	Common effects	1705.13	376				953.13
	Scalar effects	1713.35	379	8.22	3	0.042	955.35
	**Null**	**1713.4**	**380**	**0.00**	**1**	**0.084**	**953.35**
Junior high school	Common effects	1113.09	238				637.09
	Scalar effects	1116.07	241	2.98	3	0.395	634.07
	**Null**	**1116.1**	**242**	**0.00**	**1**	**0.561**	**632.07**
School adaptation							
Lower elementary school classes	Common effects	1114.53	296				522.53
	Scalar effects	1120.61	299	0.09	3	0.108	522.61
	**Null**	**1120.61**	**300**	**0.00**	**1**	**0.193**	**520.61**
Higher elementary school classes	**Common effects**	**1420.53**	**376**				**668.52**
	Scalar effects	1461.34	379	40.81	3	<0.001	703.34
	Null	1461.34	380	0.00	1	<0.001	701.34
Junior high school	**Common effects**	**921.48**	**238**				**445.48**
	Scalar effects	946.44	241	24.96	3	<0.001	464.44
	Null	946.44	242	0.00	1	<0.001	462.44
Home adaptation							
Lower elementary school classes	Common effects	1110.84	296				518.83
	Scalar effects	1119.60	299	8.77	3	0.033	521.6
	**Null**	**1119.60**	**300**	**0.00**	**1**	**0.067**	**519.6**
Higher elementary school classes	**Common effects**	**1429.01**	**376**				**677.01**
	Scalar effects	1441.05	379	12.04	3	0.007	683.05
	Null	1441.05	380	0.00	1	0.017	681.05
Junior high school	Common effects	884.62	238				408.62
	Scalar effects	887.44	241	2.82	3	0.419	405.44
	**Null**	**887.44**	**242**	**0.00**	**1**	**0.588**	**403.44**

Abbreviations: −2LL = −log-likelihood; df = degrees of freedom; diffLL = difference in likelihood; *p*-value = associated with differences in likelihood ratio between each full sex-limitation model and common effects model; AIC = Akaike’s information criterion. **Bold** font indicates the best fitting model.

**Table 6 ijerph-18-00621-t006:** Parameter estimates of *high self-appeal* from sex-limitation model fitting.

High Self-Appeal	Model	Sex	Effects of A, C, and E to Individual Differences in High Self-Appeal		
			Effects of A (95%CI)	Effects of C (95%CI)	Effects of E (95%CI)
Lower elementary school classes	Full sex-limitation	Males	0.61 (0.05–0.79)	0.00 (0.00–0.40)	0.39 (0.21–0.70)
		Females	0.16 (0.00–0.75)	0.45 (0.00–0.71)	0.39 (0.23–0.64)
	**Null model** **(Best fit model)**	**Males**	**0.60 (0.23–0.74)**	**0.00 (0.00–0.25)**	**0.40 (0.26–0.59)**
		**Females**	**0.60 (0.23–0.74)**	**0.00 (0.00–0.25)**	**0.40 (0.26–0.59)**
Higher elementary school classes	Full sex-limitation	Males	0.54 (0.05–0.69)	0.00 (0.00–0.40)	0.46 (0.31–0.66)
		Females	0.41 (0.00–0.73)	0.18 (0.00–0.63)	0.41 (0.27–0.61)
	**Null model** **(Best fit model )**	**Males**	**0.56 (0.23–0.66)**	**0.00 (0.00–0.28)**	**0.44 (0.34–0.58)**
		**Females**	**0.56 (0.23–0.66)**	**0.00 (0.00–0.28)**	**0.44 (0.34–0.58)**
Junior high school	Full sex-limitation	Males	0.17 (0.00–0.78)	0.46 (0.00–0.75)	0.37 (0.20–0.66)
		Females	0.09 (0.00–0.50)	0.09 (0.00–0.37)	0.82 (0.50–0.99)
	**Common effects model** **(Best fit model)**	**Males**	**0.12 (0.00–0.76)**	**0.50 (0.00–0.75)**	**0.38 (0.20–0.67)**
		**Females**	**0.03 (0.00–0.48)**	**0.15 (0.00–0.37)**	**0.82 (0.51–0.99)**

A = additive genetic factor, C = shared environmental factor, E = nonshared environmental factor; 95%CI = 95% confidence interval. The overlapping CI in males and females shows that parameter estimates of males and females do not significantly differ. **Bold** font indicates the best fitting model.

**Table 7 ijerph-18-00621-t007:** Parameter estimates of *sociability* from sex-limitation model fitting.

Sociability	Model	Sex	Effects of A, C, and E to Individual Differences in Socialization		
			Effects of A (95%CI)	Effects of C (95%CI)	Effects of E (95%CI)
Lower elementary school classes	Full sex-limitation	Males	0.55 (0.11–0.89)	0.27 (0.00–0.64)	0.18 (0.10–0.34)
		Females	0.69 (0.21–0.90)	0.14 (0.00–0.57)	0.17 (0.10–0.31)
	**Null model** **(Best fit model)**	**Males**	**0.50 (0.23–0.80)**	**0.32 (0.03–0.55)**	**0.18 (0.12–0.27)**
		**Females**	**0.50 (0.23–0.80)**	**0.32 (0.03–0.55)**	**0.18 (0.12–0.27)**
Higher elementary school classes	Full sex-limitation	Males	0.21 (0.00–0.68)	0.52 (0.07–0.75)	0.26 (0.17–0.40)
		Females	0.84 (0.38–0.90)	0.52 (0.00–0.45)	0.16 (0.10–0.26)
	**Null model** **(Best fit model)**	**Males**	**0.59 (0.31–0.84)**	**0.21 (0.48–0.46)**	**0.21 (0.15–0.28)**
		**Females**	**0.59 (0.31–0.84)**	**0.21 (0.48–0.46)**	**0.21 (0.15–0.28)**
Junior high school	Full sex-limitation	Males	0.81 (0.39–0.89)	0.00 (0.00–0.40)	0.19 (0.11–0.34)
		Females	0.65 (0.04–0.79)	0.09 (0.00–0.54)	0.35 (0.21–0.57)
	**Null model** **(Best fit model)**	**Males**	**0.73 (0.35–0.81)**	**0.00 (0.00–0.34)**	**0.27 (0.19–0.40)**
		**Females**	**0.73 (0.35–0.81)**	**0.00 (0.00–0.34)**	**0.27 (0.19–0.40)**

A = additive genetic factor, C = shared environmental factor, E = nonshared environmental factor; 95%CI = 95% confidence interval. The overlapping CI in males and females shows that the parameter estimates of males and females do not significantly differ. **Bold** font indicates the best fitting model.

**Table 8 ijerph-18-00621-t008:** Parameter estimates of *school adaptation* from sex-limitation model fitting.

School Adaptation	Model	Sex	Effects of A, C, and E to Individual Differences in School Adaptation		
			Effects of A (95%CI)	Effects of C (95%CI)	Effects of E (95%CI)
Lower elementary school classes	Full sex-limitation	Males	0.84 (0.51–0.91)	0.00 (0.00–0.25)	0.16 (0.09–0.25)
		Females	0.16 (0.00–0.77)	0.52 (0.00–0.75)	0.32 (0.18–0.52)
	**Null model** **(Best fit model)**	**Males**	**0.77 (0.45–0.84)**	**0.00 (0.00–0.27)**	**0.23 (0.16–0.35)**
		**Females**	**0.77 (0.45–0.84)**	**0.00 (0.00–0.27)**	**0.23 (0.16–0.35)**
Higher elementary school classes	Full sex-limitation	Males	0.49 (0.25–0.87)	0.45 (0.06–0.68)	0.68 (0.04–0.11)
		Females	0.29 (0.00–0.74)	0.34 (0.00–0.66)	0.37 (0.24–0.57)
	**Common effects model** **(Best fit model)**	**Males**	**0.49 (0.25–0.93)**	**0.44 (0.00–0.68)**	**0.07 (0.04–0.11)**
		**Females**	**0.63 (0.00–0.76)**	**0.00 (0.00–0.67)**	**0.36 (0.24–0.54)**
Junior high school	Full sex-limitation	Males	0.95 (0.78–0.97)	0.00 (0.00–0.16)	0.05 (0.00–0.11)
		Females	0.54 (0.32–0.69)	0.00 (0.00–0.18)	0.46 (0.31–0.65)
	**Full sex-limitation model** **(Best fit model)**	**Males**	**0.95 (0.78–0.97)**	**0.00 (0.00–0.16)**	**0.05 (0.03–0.11)**
		**Females**	**0.54 (0.32–0.69)**	**0.00 (0.00–0.18)**	**0.46 (0.31–0.65)**

A = additive genetic factor, C= shared environmental factor, E= nonshared environmental factor; 95%CI = 95% confidence interval. The overlapping CI in males and females shows that the parameter estimates of males and females do not significantly differ. **Bold** font indicates the best fitting model.

**Table 9 ijerph-18-00621-t009:** Parameter estimates of *home adaptation* from sex-limitation model fitting.

Home Adaptation	Model	Sex	Effects of A, C, and E to Individual Differences in Home Adaptation		
			Effects of A (95%CI)	Effects of C (95%CI)	Effects of E (95%CI)
Lower elementary school classes	Full sex-limitation	Males	0.00 (0.00–0.29)	0.81 (0.54–0.88)	0.19 (0.11–0.30)
		Females	0.44 (0.09–0.83)	0.36 (0.00–0.67)	0.19 (0.11–0.35)
	**Null model** **(Best fit model)**	**Males**	**0.36 (0.09–0.63)**	**0.44 (0.19–0.64)**	**0.20 (0.13–0.31)**
		**Females**	**0.36 (0.09–0.63)**	**0.44 (0.19–0.64)**	**0.20 (0.13–0.31)**
Higher elementary school classes	Full sex-limitation	Males	0.37 (0.00–0.78)	0.40 (0.01–0.72)	0.23 (0.15–0.34)
		Females	0.21 (0.02–0.51)	0.69 (0.39–0.85)	0.10 (0.06–0.17)
	**Common effects model** **(Best fit model)**	**Males**	**0.35 (0.00–0.66)**	**0.43 (0.13–0.71)**	**0.23 (0.15–0.35)**
		**Females**	**0.21 (0.02–0.50)**	**0.69 (0.40–0.85)**	**0.10 (0.06–0.17)**
Junior high school	Full sex-limitation	Males	0.50 (0.11–0.89)	0.32 (0.00–0.67)	0.17 (0.09–0.34)
		Females	0.82 (0.41–0.95)	0.09 (0.00–0.49)	0.09 (0.05–0.17)
	**Null model** **(Best fit model)**	**Males**	**0.53 (0.29–0.83)**	**0.34 (0.04–0.56)**	**0.13 (0.09–0.21)**
		**Females**	**0.53 (0.29–0.83)**	**0.34 (0.04–0.56)**	**0.13 (0.09–0.21)**

A = additive genetic factor, C = shared environmental factor, E = nonshared environmental factor; 95%CI = 95% confidence interval. The overlapping CI in males and females shows that the parameter estimates of males and females do not significantly differ. **Bold** font indicates the best fitting mode. For high self-appeal, in elementary school groups, 56% to 60% heritability was found, with nonshared environmental influences accounting for 40% to 44% of variance. Sex difference only exists in junior high school. In total, 12% heritability was found, with shared environment accounting for 50% of variance and nonshared environmental influences accounting for 38% of variance in males, versus 3% heritability, 15% shared environmental influences, and 82% nonshared environmental influences on variance in females.

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
