# Peer review of "The Differential Heritability of Social Adjustment by Sex"

_ijerph, 2021, doi:10.3390/ijerph18020621_

Round 1

Reviewer 1 Report

The current manuscript attempts at identifying the potential differences across genders on the influence of genetic and environmental (both shared and unshared) factors on social adjustment. Results show that there seems to be gender differences at distinct developmental phases (as evaluated by age groups) for most of the inspected social adjustment constructs.

  • Broad comments:
    • [General] The major drawback of the paper is that it is clear in some aspects (or sections) and confusing in others, giving  the sensation that the authors have written their parts independent of each other and then conjoined these sections, but no one has proofread the entire manuscript to be sure of its coherence and cohesion. Fortunately, the research question seems novel, relevant and interesting, and the results comprehensive and rigorous. Furthermore, the potential to achieve a paper that can be understood and informative by the readers of a rather broad scope journal as this one is present, since some parts are already very adequate. The most important features to change in the current manuscript are to include some structure and to elaborate some key aspects trying to be “didactic” to make the paper more accessible. This modification shall be made for the Introduction, Methods, Results and Discussion sections.
    • The [Introduction] section needs to be honed. The aim and hypothesis paragraph is somewhat decontextualized and does not seem to entirely follow the previous information. In other words, the Introduction is insufficient to fully understand the objective of the work. Additionally, it seems odd to read the hypothesis before the aim of the study since usually the hypothesis is derived from the objective in conjunction with the information previously presented in the rest of the section. Thus, it is advised that the information provided in the introductory paragraphs about prior results found in relation to genetic influences, gender and social adjustment will be further elaborated. It would also help the reader to make an attempt to generate an argumentative arc that makes all the paragraphs more cohesive, so that they do not seem so disconnected from each other (especially the first two).
    • [Data analysis subsection + Results section] Although both the Methods and Results sections are well written in general, these two sections (regarding the Methods sections, the following only happens for the Data analysis subsection) can be sometimes confusing because there is a lack of structure that helps the reader identify what is the rationale behind some of the analyses conducted and what to expect (or learn) from them. In this vein, and to aggravate the situation, there are a lot of different comparisons between different models. For example, what is the difference between the results of Tables 4, 5 and 6 to 9, and what is their distinct relevance? Therefore, it would be desirable to include in the Data analysis subsection which are the most relevant analyses that will be found in the Results section and add a tad of information about them (in some cases, this information is already available in the current manuscript in form of cites, yet it remains unexplained unless the references are read). This information should include what are the procedures (and the name of the algorithms used), why are the procedures used and what information will they provide (this last aspect is well covered for model comparisons, but not for ICC, for example). Moreover, in order to make it easier to keep track of the results, it would be helpful to divide the Results section into different subsections (according to what has been previously done in the data analysis section). In this way, information would be more clear and easy to find in both sub/sections.
    • [Discussion] There is little discussion. This section’s content is mainly comprised by repeating the results and providing some explanation with few links to other prior related (to some degree if this has not thoroughly been studied before) research that support the explanation (which is tentative/speculative and without proposing plausible alternatives in many cases). Including a more exhaustive and elaborated discussion (where suitable) would be appreciated and would help grasping the relevance and novelty of the outcomes of the study and their actual implications for either, or both, theory and practice (beyond the two final lines).
  • Specific comments:
    • [General] ‘Sex’ tends to be more genetic-related and ‘gender’ more sociological-related. In this study, there does not seem to be the possibility for participants to report their gender identity; they are categorized as male or female according to a “genetic-like” way (e.g., medical records?). Then, why has the word ‘gender’ (instead of ‘sex’) been chosen?
    • [page1, lines 38-39] For the readers’ sake (at least for those less familiar with heritability), it would be advisable to maintain the 'percentage format' that has been used throughout the same paragraph, instead of the 'proportion'. Otherwise, it could be understood as another measure. Moreover, in the original paper, the value reported (.39) seems to reflect the total effect size, instead of the twins study design (i.e., .47). Since the paragraph of the present study seems to talk about twin studies, this aspect could be clarified.
    • [page2, line 12] Is it ‘founded’ or ‘found’?
    • [page2, line 15] The word ‘heritability’ seems to be written in a different font size.
    • [Participants subsection] Authors claim their sample is comprised by 467 twin pairs. However, when they provide the specific numbers according to their classification, these sums a total of 468 twin pairs (regardless of the category). Please, check (and correct where suitable) the total or specific number of twin pairs to make the numbers match.
    • [Participants subsection] Is there any information regarding whether all the twin pairs of the sample shared the same environment (family, school…) or not? Was this simply assumed to be true? If available, please add this information. If not, make it acknowledgeable.
    • [Procedure subsection] The 'previous study' (page 2, line 34) is not cited/referenced.
    • [Instruments subsection] Since, apparently, the questionnaire is only available in Japanese, some more information about it would be appreciated. Even though some aspects are treated in the Limitations section, previous additional information would be helpful, especially in regards to the high self-appeal construct (which is apparently the most novel scale, and for which the brief definition provided in the aims section and the items do not seem to fully concur).
    • [Figure 1] The in-figure captions of panels B and C are misplaced.
    • [Table 1] It would be interesting to see the questionnaires’ scales descriptives broken by age group as well as the total.
    • [Tables 6 to 9] Please, adjust the columns’ width in order to make them similar to Table 7, which seems to be the only one displaying all values correctly.

Author Response

Response: We thank you for your feedback and helpful suggestions. We have revised the paper significantly in response to the reviewers’ comments and we hope the revised version is acceptable for publication. We have highlighted the revised text in green in the main manuscript. Please let us know if any further modifications are required.

Response to Broad comments

We have rewritten our manuscript are to include some structure.

[Introduction]

We have rewritten our introduction section to be honed.

・We added the first paragraph about the current situation in Japan.(p1.line28-31)

・We reviewed the benchmark studies as follows.(p1. Line31-33)

“Twin study reported that shared environmental effects (40%) were found to play a moderate role for symptoms of separation anxiety and the parameter approached significance among 11 to 13-year-old males [5].”

・We switched the order of hypothesis and objective.(p2. Line25-29)

[Data analysis and Results]

・The Data analysis section is now divided into subsections.

 2.5. Data analysis

2.5.1. Model fitting of twin studies methods

2.5.2 Qualitative sex differences with Sex limitation model fitting

2.5.3 Quantitative sex differences with Sex limitation model fitting

2.5.4. Interpretation of the results of model comparisons

・The Results section is now divided into subsections.

  1. Results

3.1 Descriptive statistics

3.2 Sex limitation model fitting

3.2.1 Qualitative sex differences

3.2.2 Quantitative sex differences

・We changed the titles of the tables to clarify the differences in the resulting tables.

Table 1. The Mean and Standard Deviation for Measures: One-way analysis (N=467).

Table 2. Bivariate Correlation Matrix: Pearson (N=467).

Table 4. Comparison of full sex-limitation models and common effects models to consider qualitative sex differences

Table 5. Comparison of common effects, scalar effects models and null models to consider quantitative sex difference.

・We add a information about results in tables.

Effects of A, C, and E to individual differences in high self-appeal (Table6)

Effects of A, C, and E to individual differences in sociability (Table7)

Effects of A, C, and E to individual differences in school adaptation (Table8)

Effects of A, C, and E to individual differences in home adaptation (Table9)

[Discussion]

We have rewritten our discussion section.

・We added this section.

  “One reason for the high genetic effect on school adjustment among junior high school students may be the influence of the junior high school entrance exam. Although this study did not investigate whether twins attend the same school, more twins will likely attend separate schools starting in middle school, especially male and female twins. For example, boys may go to a boys’ school and girls to a girls’ school. This may have increased the difference between male and female twins in school adjustment, which may have affected the results.” (p13, line55-60)

“Twins were often compared with those around them from birth, and self-appeal or adaptation to group styles may be different from that of singletons. This research helps us to consider not only genetic and environmental variances in social adjustment, but also how schoolchildren and adolescents adapt to their environment.” (p14, line93-96)

Response to Specific comments

[General] ‘Sex’ tends to be more genetic-related and ‘gender’ more sociological-related. In this study, there does not seem to be the possibility for participants to report their gender identity; they are categorized as male or female according to a “genetic-like” way (e.g., medical records?). Then, why has the word ‘gender’ (instead of ‘sex’) been chosen?

Response: Thank you for your meaningful comments. We corrected gender to sex.

[page1, line38-39]

For the readers’ sake (at least for those less familiar with heritability), it would be advisable to maintain the 'percentage format' that has been used throughout the same paragraph, instead of the 'proportion'. Otherwise, it could be understood as another measure. Moreover, in the original paper, the value reported (.39) seems to reflect the total effect size, instead of the twins study design (i.e., .47). Since the paragraph of the present study seems to talk about twin studies, this aspect could be clarified.

Response: Thank you for your meaningful comments. We have revised the manuscript as follows. “According to the meta-analysis, the answer is 40% average effect to genetic contributions to individual differences in personality [15].” (page2, line1-2)

[page2, line 12] Is it ‘founded’ or ‘found’?

Response: We have revised the manuscript to “found”. (page1, line39)

[page2, line 15] The word ‘heritability’ seems to be written in a different font size.

Response: We have revised the font size. (page2, line24)

[Participants subsection] Authors claim their sample is comprised by 467 twin pairs. However, when they provide the specific numbers according to their classification, these sums a total of 468 twin pairs (regardless of the category). Please, check (and correct where suitable) the total or specific number of twin pairs to make the numbers match.

Response: Thank you for your meaningful comments. We have revised the manuscript as follows.

82 DZ female (DZf) were collect. (p2, line35-36)

[Participants subsection] Is there any information regarding whether all the twin pairs of the sample shared the same environment (family, school…) or not? Was this simply assumed to be true? If available, please add this information. If not, make it acknowledgeable.

Response: Thank you for your meaningful comments. We have revised the limitation as follows.

“Last, because we did not investigate whether twins attend the same school, we could not consider its impact on school adjustment.” (p14, line108-109)

[Procedure subsection] The 'previous study' (page 2, line 34) is not cited/referenced.

Response: Thank you for your meaningful comments. We have added the manuscript as follows.

“In our previous study, 14 pairs of twins were excluded from the study because of cerebral palsy, cleft palate, autism or Down’s syndrome, to remove influential outliers [24].” (p3, line1)

[Instruments subsection] Since, apparently, the questionnaire is only available in Japanese, some more information about it would be appreciated. Even though some aspects are treated in the Limitations section, previous additional information would be helpful, especially in regards to the high self-appeal construct (which is apparently the most novel scale, and for which the brief definition provided in the aims section and the items do not seem to fully concur).

Response: Thank you for your meaningful comments. We have revised the manuscript as follows.

“In our previous study, 14 pairs of twins were excluded from the study because of cerebral palsy, cleft palate, autism or Down’s syndrome, to remove influential outliers [24].” (p3, line1)

[Figure 1] The in-figure captions of panels B and C are misplaced.

Response: We have replaced the captions of panels B and C.

[Table 1] It would be interesting to see the questionnaires’ scales descriptives broken by age group as well as the total.

Response: Thank you for your meaningful comments. We have revised the table 1 as follows.

Measure  

The lower classes

(N=152)

The higher classes

(N=192)

Junior high school

(N=123)

p-value

Multiple-comparison

(Bonftrroni)

Mean±SD

Mean±SD

Mean±SD

(1) High self-appeal

3.35±2.32

3.10±2.29

1.96±1.69

<0.001

Between lower class and junior high school, between higher classes and junior high school

(2) Sociability

2.57±2.40

2.62±2.49

3.30±2.54

0.023

n.s.

(3) School adaptation

1.28±1.54

1.27±1.73

1.65±1.63

0.111

n.s.

(4) Home adaptation

2.73±1.67

2.56±1.88

2.16±1.78

0.043

n.s.

[Tables 6 to 9] Please, adjust the columns’ width in order to make them similar to Table 7, which seems to be the only one displaying all values correctly.

Response: Thank you for your meaningful comments. We have redisplaying all values correctly.

Reviewer 2 Report

I think your study is a very meaningful study trial for twins. However, from a few perspectives, corrections are needed. 

First, there are limitations in the validity and reliability of research tools.

Second, basic correction is required when creating a table.

Third, research result part. When describing statistical results, you should follow the principle of describing by following the basic description format that there is a significant difference in general.

When describing the statistical result, it is necessary to accurately confirm it and then describe it.

Line 16 : “high-self appeal was not related to school adaptation. “ (incorrect)

Line 17 : “In addition, sociability was positively related to sociability”  (incorrect)

Table 1.  describe total number. 

Table 2. describe total number.

Please write the name of the statistics at the top of the table. Ex ) r (p value)

Describe the Footnote .  

Table 3.  describe total number.

Please write the name of the statistics at the top of the table. Ex ) r (p value)

Thank you.

Author Response

Response: We thank you for your feedback and helpful suggestions. We have revised the paper significantly in response to the reviewers’ comments and we hope the revised version is acceptable for publication. We have highlighted the revised text in green in the main manuscript. Please let us know if any further modifications are required.

■First, there are limitations in the validity and reliability of research tools.

Response: Thank you for your comments. We added the limitation as follows.

“There are several limitations in our study. First, this study used a parent-reporting questionnaire on twins’ social ability. Parent ratings of children’s personality are affected by contrast effects that exaggerate estimates of genetic influence in twin studies. Second, while the legitimacy and reliability of the scales employed in this research have been verified, its scope is limited by the fact that its scales were created in Japan and have not been used in international comparisons.” (p14, line98-103)

■Second, basic correction is required when creating a table.

Response: Thank you for your meaningful comments. We have redisplaying tables all values correctly.

■Third, research result part. When describing statistical results, you should follow the principle of describing by following the basic description format that there is a significant difference in general.

Response: Thank you for your meaningful comments. We have revised the manuscript as follows.

“One-way analysis indicated significant differences in high self-appeal (p<0.001), sociability (p=0.023), and home adaptation (p=0.043). There were significant differences between the lower classes and junior high school in high self-appeal, between the higher classes and junior high school with multiple comparisons (Bonferroni). Table 2 shows the bivariate correlation matrix (Pearson). At the bivariate level, there was a significant correlation between sociability and school adaptation (p<0.001), between high self-appeal and home adaptation (p<0.001), and between sociability and home adaptation (p<0.001). High-self appeal was not related to sociability and school adaptation.” (p6, line26-33).

■When describing the statistical result, it is necessary to accurately confirm it and then describe it. Line 16 : “high-self appeal was not related to school adaptation. “ (incorrect). Line 17 : “In addition, sociability was positively related to sociability”  (incorrect)

Response: Thank you for your meaningful comments. There was an error in the figures in the table. We have revised the table 1 as follows.

Measure

(1)

(2)

(3)

(4)

r=

r=

r=

r=

(1) High self-appeal

1

-0.094

0.065

0.432**

(2) Sociability

1

0.707**

0.210**

(3) School adaptation

1

0.330

(4) Home adaptation

1

■Table 1.  describe total number. 

Response: Thank you for your meaningful comments. We added total number in table1 as follows.

Table 1. The Mean and Standard Deviation for Measures: One-way analysis (N=467).

■Table 2. describe total number. Please write the name of the statistics at the top of the table. Ex ) r (p value) Describe the Footnote.

Response: Thank you for your meaningful comments. We added total number in table2 as follows.

Table 2. Bivariate Correlation Matrix: Pearson (N=467).

Measure

(1)

(2)

(3)

(4)

r=

r=

r=

r=

(1) High self-appeal

1

-0.094

0.065

0.432**

(2) Sociability

1

0.707**

0.210**

(3) School adaptation

1

0.330

(4) Home adaptation

1

■Table 3.  describe total number. Please write the name of the statistics at the top of the table. Ex ) r (p value)

Response: Thank you for your meaningful comments. We added total number and the name of the statics at the top of table3.

                     Zygosity & Sex (N=467)

            Intra-class Correlations

MZm

MZf

DZm

DZf

DZos

(N=101)

(N=96)

(N=87)

(N=82)

(N=101)

r=

r=

r=

r=

r=

Measure

High self-appeal

The lower classes

0.71

0.55

0.18

0.51

0.21

The higher classes

0.59

0.66

0.17

0.34

0.10

Junior high school

0.36

0.14

0.60

0.18

0.16

Sociability

The lower classes

0.89

0.81

0.56

0.41

0.62

The higher classes

0.74

0.88

0.68

0.28

0.41

Junior high school

0.86

0.72

0.22

0.37

0.27

School adaptation

The lower classes

0.91

0.66

0.22

0.62

0.27

The higher classes

0.9

0.77

0.77

0.37

0.37

Junior high school

0.9

0.51

0.16

0.1

0.78

Home adaptation

The lower classes

0.68

0.79

0.83

0.57

0.52

The higher classes

0.82

0.91

0.56

0.79

0.55

Junior high school

0.72

0.88

0.66

0.61

0.70

Round 2

Reviewer 2 Report

Thank you for your correction.